# Negative Charge-Carrying Glycans Attached to Exosomes as Novel Liquid Biopsy Marker

**DOI:** 10.3390/s24041128

**Published:** 2024-02-08

**Authors:** Natalia Kosutova, Lenka Lorencova, Michal Hires, Eduard Jane, Lubomir Orovcik, Jozef Kollar, Katarina Kozics, Alena Gabelova, Egor Ukraintsev, Bohuslav Rezek, Peter Kasak, Hana Cernocka, Veronika Ostatna, Jana Blahutova, Alica Vikartovska, Tomas Bertok, Jan Tkac

**Affiliations:** 1Institute of Chemistry, Slovak Academy of Sciences, Dubravska cesta 5807/9, 845 38 Bratislava, Slovakialenka.lorencova@savba.sk (L.L.); eduard.jane@savba.sk (E.J.);; 2Institute of Materials and Machine Mechanics, Slovak Academy of Sciences, Dubravska cesta 9/6319, 845 13 Bratislava, Slovakia; 3Polymer Institute, Slovak Academy of Sciences, Dubravska cesta 9, 845 41 Bratislava, Slovakia; 4Biomedical Research Centre, Slovak Academy of Sciences, Dubravska cesta 9, 845 05 Bratislava, Slovakiaalena.gabelova@savba.sk (A.G.); 5Department of Physics, Faculty of Electrical Engineering, Czech Technical University in Prague, Technicka 2, 166 27 Prague, Czech Republic; ukraiego@fel.cvut.cz (E.U.); rezekboh@fel.cvut.cz (B.R.); 6Centre for Advanced Materials, Qatar University, Doha P.O. Box 2713, Qatar; peter.kasak@qu.edu.qa; 7Institute of Biophysics, Czech Academy of Sciences, Kralovopolska 135, 61200 Brno, Czech Republic; cernocka@ibp.cz (H.C.);

**Keywords:** exosomes, prostate cancer, surface plasmon resonance, self-assembled monolayer, nanoparticle tracking analysis, microscopy techniques

## Abstract

Prostate cancer (PCa) is the second most common cancer. In this paper, the isolation and properties of exosomes as potential novel liquid biopsy markers for early PCa liquid biopsy diagnosis are investigated using two prostate human cell lines, i.e., benign (control) cell line RWPE1 and carcinoma cell line 22Rv1. Exosomes produced by both cell lines are characterised by various methods including nanoparticle-tracking analysis, dynamic light scattering, scanning electron microscopy and atomic force microscopy. In addition, surface plasmon resonance (SPR) is used to study three different receptors on the exosomal surface (CD63, CD81 and prostate-specific membrane antigen-PMSA), implementing monoclonal antibodies and identifying the type of glycans present on the surface of exosomes using lectins (glycan-recognising proteins). Electrochemical analysis is used to understand the interfacial properties of exosomes. The results indicate that cancerous exosomes are smaller, are produced at higher concentrations, and exhibit more nega tive zeta potential than the control exosomes. The SPR experiments confirm that negatively charged *α*-2,3- and *α*-2,6-sialic acid-containing glycans are found in greater abundance on carcinoma exosomes, whereas bisecting and branched glycans are more abundant in the control exosomes. The SPR results also show that a sandwich antibody/exosomes/lectins configuration could be constructed for effective glycoprofiling of exosomes as a novel liquid biopsy marker.

## 1. Introduction

All living organisms share a common characteristic: their surface is covered by glycans. Glycosylation is a process by which one or more monosaccharide or oligosaccharide units are linked to proteins or lipids; this is an essential co- and post-translational modification of proteins [1,2]. This process is controlled by a series of glycosyltransferases and glycosidases found in the endoplasmic reticulum and a Golgi apparatus. Glycans—complex carbohydrates covalently attached to proteins or lipids—are a part of various glycopeptides and glycoproteins, peptidoglycans, glycolipids, lipopolysaccharides, glycosaminoglycans and other glycosides [3]. There are two major protein glycosylation types, *N*-glycosylation represented by *N*-glycans attached to the amine group of the amino acid side-chain within the Asn-X-Ser/Thr sequence (X ≠ Pro) and *O*-glycosylation, which, on the other hand, is represented by *O*-glycans linked to the oxygen atom of several amino acid residues, including serine and threonine [4]. Glycans are involved in a wide range of key cellular processes, including intra- and intercellular signalling, cell growth and differentiation or immune response and, due to their size and hydrophilicity, glycan chains can significantly alter the physicochemical properties of glycoproteins, such as solubility, stability (half-life), viscosity or charge [5,6,7]. A change in the glycosylation pattern has been associated with mechanisms underlying tumourigenesis and metastasis for various types of tumour progression, including the most common ones, such as breast, colorectal or prostate carcinomas [8,9,10]. These are some of the reasons why interest in the study of glycans has been steadily increasing in recent decades with an increasing focus on the study of glycosylation of exosomes.

Exosomes are naturally produced biological (nano)particles in a range of sizes from 30 nm to 200 nm [11]. They are produced by cells and released into various body fluids, including blood, lymph, urine, cerebrospinal fluid, bile, saliva or breast milk. Exosomes are formed when multi-vesicular bodies fuse with a plasma membrane [12,13]. They are carriers of different types of biomolecules, such as lipids, proteins and nucleic acids, and play an important role in intercellular communication. The way they are formed, their composition, low immunogenicity and non-toxicity make them an interesting subject of investigation not only for diagnostic but also for prognostic and therapeutic purposes. Their release increases dramatically in cases of tumourigenesis, which opens up the possibility of their application as a new source of biomarkers [14]. 

In general, the analysis of biomarkers is challenging due to their low levels in serum, the higher degree of complexity of potential analytes and the limitations of current analytical methods [15] with some advanced techniques for their analysis emerging [16]. Biomarkers may be produced by the cancer tissue itself or by other cells in the body in response to cancer. For most diagnostic approaches in oncology, imaging tests and, finally, biopsies are the gold standard, although so-called liquid biopsy approaches are currently on the rise [11]. Liquid biopsy based on the analysis of biofluids is a sensitive and non-invasive method applicable for diagnosis. Liquid biopsy is minimally invasive and requires only a small amount of a sample. It could provide diagnostic, prognostic and/or therapeutic information and it could be used for monitoring metastases, identifying changes in genetic or epigenetic codes, determining optimal treatments for individual patients and assessing treatment efficacy [17]. Aberrant glycoform profiling may also be used as a novel liquid biopsy approach in combination with tissue-specific glycoproteins (as in the case of the free prostate-specific antigen (free PSA for prostate cancer) or extracellular vesicles, such as exosomes [18].

Prostate cancer (PCa) is the second most common cancer, with almost 1.4 million new cases and 375,000 deaths worldwide in 2020, and is the most frequently diagnosed cancer in men in more than half of the world’s countries. The incidence is predicted to grow by ~35% by 2030 [19]. Commonly used biomarkers to detect PCa early involve PSA and its variations, such as total PSA, free PSA (%), PSA velocity and PSA density, or more advanced methods, such as the Prostate Health Index (by Beckman Coulter) or 4 k Score (OPKO Health) tests [20]. Although the discoverer of PSA itself, Dr. Richard Ablin, claimed that PSA was never meant to be used as a diagnostic tool and has been widely misused, its main advantage is undoubtedly its tissue specificity, which is unique among all FDA-approved biomarkers. Alternatives to PSA-testing are still regarded as an unmet medical need to help avoid any unnecessary prostate biopsies or to identify unproven low-grade tumours using multiparametric magnetic resonance imaging (mpMRI) [21,22,23]. The role of exosomes in today’s diagnostic algorithm is examined in this study. Different approaches to using exosomes as a rich source of biomarkers are currently present in the literature, while many barriers to their clinical use still exist—as a diagnostic tool but also as a novel drug-delivery system [24]. One of these is clearly the high biological variability in their size and quality as a result of the isolation procedure chosen and the storage conditions [25]. Secondly, clinically relevant information needs to be obtained by real-life population studies (currently, only studies using biological models are performed), and the whole process of analysis (including the pre-analytical phase involving centrifugation, etc.) needs to present a user-friendly platform compatible with common clinical practice [26]. ELISA-based methods presented by Logozzi et al. [27,28] are the ideal solution to overcome the latter in the case of prostate cancer. In our previous study, we thoroughly investigated different isolation approaches as well as exosomes’ stability at different temperatures, with the aim being to analyse intact exosomes compared to only fragments or biomarkers encapsulated inside these vesicles [15]. For this analysis, an interfacial effect also needs to be taken into account. For SPR experiments and the glycoprofiling of the intact exosomes described further in the text, the surface charge was investigated as a main driving force in the immobilisation process and a differentiator between exosomes produced by 22Rv1 (cancerous) and RWPE1 (benign) prostate cell lines using different surface modification of SPR gold sensors. Hence, this study aims to further accelerate the current exploration stage in order to remove these blocks in exosome applications.

## 2. Materials and Methods

### 2.1. Chemicals

All commonly used chemicals, e.g., buffer components (phosphate buffer—PB, consisting of K_2_HPO_4_ and KH_2_PO_4_), phosphate saline buffer (PBS) tablets or basic inorganic and organic compounds (such as 11-mercaptoundecanoic acid (MUA), 6-mercapto-1-hexanol (MCH) or bovine serum albumin (BSA), human serum albumin (HSA) 30% aqueous solution, trehalose, 4-(2-hydroxyethyl)-1-piperazineethanesulphonic acid (HEPES)) were supplied by Sigma-Merck (St. Louis, MO, USA). PB-HAT buffer was prepared by mixing PB, 20 mM HEPES, 0.2% HSA and 25 mM trehalose. All buffer solutions were freshly prepared in 0.055 μS ultrapure deionised water (DW) and filtered *prior* to use using 0.22 μm sterile filters. The anti-CD63 (ab1318), anti-CD81 (ab286173) and anti-PSMA (ab268061) monoclonal antibodies were supplied by Abcam (Cambridge, UK). All SPR procedures, i.e., pH immobilisation-scouting (using acetate buffers with pH 4.0, 4.5, 5 and 5.5), amine-coupling (using EDC, NHS and ethanolamine solutions) and single-cycle kinetics (SCK) analyses (kinetic titrations, including chip regeneration using 10 mM glycine HCl, pH 2.5) were performed using kits supplied by GE Healthcare (Vienna, Austria). The following unconjugated forms of lectins for the SPR analysis were provided by Vector Laboratories (USA): *Aleuria aurantia* lectin (AAL, L-1390), *Galanthus nivalis* lectin (GNL, L-1240), *Maackia amurensis* agglutinin II (MAAII, L-1260), *Phaseolus vulgaris* erythroagglutinin (PHA-E, L-1120), *Phaseolus vulgaris* leucoagglutinin (PHA-L, L-1110), *Pisum sativum* agglutinin (PSA, L-1050), *Sambucus nigra* agglutinin (SNA, L-1300) and *Wisteria floribunda* agglutinin (WFL, L-1350). To regenerate the biorecognition surface and remove the attached lectins, elution solutions from Vector Laboratories (Burlingame, CA, USA) were purchased, i.e., glycoprotein-eluting solution for mannose/glucose binding lectins (ES-1100-100); glycoprotein-eluting solution for galactose/GalNAc binding lectins (ES-2100-100); glycoprotein-eluting solution for fucose/arabinose binding lectins (ES3100-100); glycoprotein eluting solution for GlcNAc/Chitin binding lectins (ES-5100-100); glycoprotein-eluting solution for sialic acid binding lectins (ES-7100-100).

### 2.2. Cell Culture Cultivation

The human benign prostatic epithelial RWPE1 cell line (ATCC CRL-11609) was cultivated in a Keratinocyte serum-free medium (Cat. No. 17005–042) supplemented with BPE (bovine pituitary extract) and EGF (epidermal growth factor) provided with the medium and antibiotics (penicillin 100 U·mL^−1^, streptomycin 100 μg·mL^−1^). The human prostate carcinoma 22Rv1 cell line (ATCC CRL-2505) was cultured in an ATCC-formulated Roswell Park Memorial Institute (RPMI)-1640 medium (ATCC 30–2001) supplemented with 10% exosome-depleted foetal bovine serum (A2720801) and antibiotics (penicillin 100 U·mL^−1^, streptomycin 100 μg·mL^−1^). Cell lines were purchased from the American Type of Culture Collection (ATCC, Manassas, VA, USA). The cells were maintained at 37 °C in a humidified atmosphere of 5% CO_2_. The culture media, foetal bovine serum (FBS), antibiotics and other chemicals used for cell cultivation were purchased from GIBCO (Thermo Fisher Scientific, Waltham, MA, USA).

### 2.3. Exosome Collection and Isolation

From the media collected after the RWPE1 or 22Rv1 cell lines’ cultivation, the samples were obtained through centrifugation at 10,000× *g* at 4 °C for 30 min to collect a free supernatant without larger vesicles (microvesicles or apoptotic bodies). This supernatant was further used for exosome collection and analysis using nanoparticle-tracking analysis and stored at −80 °C unless indicated otherwise. For the purposes of isolation, a MagCapture™ exosome isolation kit (FUJIFILM, Wako Chemicals, Richmond, VA, USA) was used. The exosome isolation technique was chosen based on the criteria and experiments described in our previous study [15]. Exosomes were always freshly isolated *prior* to each SPR analysis to avoid any effect of short- or long-term storage on the experiment results. In the event of their storage, a ReadyShield^®^ protease inhibitor cocktail (Sigma-Merck, USA) was used according to the manufacturer’s recommendations.

### 2.4. Nanoparticle-Tracking Analysis (NTA)

Samples containing exosomes were analysed using a NanoSight NS300 Instrument (NanoSight, Malvern, UK). Exosomes were prepared in a concentration range from 10^7^ *per* mL up to 10^9^ *per* mL to ensure the accuracy of NTA in accordance with the manufacturer’s recommendations. The samples were diluted in 10 mM PB buffer pH 7.4. The same buffer solution was used to clean the chamber between measurements together with 70% ethanol. Software settings for the entire analysis were as follows: detection threshold: 5; blue 488 nm laser beam was applied to the loaded sample; the video was recorded three times, for 45 s at a frame rate of 25 frames·s^−1^. The video was analysed using NTA v.3.4 software. The hydrodynamic diameter of the particles in a suspension was calculated following the two-dimensional Stokes–Einstein equation [29]. After the video was analysed, the sizes and concentrations of exosomes were obtained and related to the cell count in a production bioreactor. Statistical evaluation was performed using Microsoft Excel 16.0.17231.20194.

### 2.5. Dynamic Light-Scattering (DLS)

Zeta potential and size measurements were performed using Zetasizer Nano-ZS (Malvern Instruments, Malvern, UK) equipped with a helium/neon laser (λ = 633 nm) and thermo-electric temperature controller at a scattering angle of 173° at 25 °C in the disposable cuvette and in the zeta cell, respectively. The intensity-averaged particle diameter and the polydispersity index values were calculated from the cumulants analysis. For the measurements, each sample was diluted with PB-HAT buffer pH 7.0.

### 2.6. Microscopic Analysis (SEM and AFM)

For the scanning electron microscopy analysis (SEM), samples were diluted 10k times in PB. The silicon substrates were ultrasonically cleaned with ethanol and subsequently dried with nitrogen stream prior to dropcasting. 3.5 µL of samples were applied on silicon substrates for better conductivity of sample The samples were observed via SEM JEOL JSM-7600F at the acceleration voltage of 10 kV for reducing of charging of samples with aperture 30 nm for higher resolution and with use of Low Angle BE Detector (LABE) for better observation of exosomes.

For the correlative probe-electron microscopy analysis (CPEM) [30,31], combining in-situ Atomic Force Microscopy (AFM) and Scanning Electron Microscopy (SEM), exosome samples were diluted 500 times in PB and 3.5 µL was applied via drop-casting on p-type silicon substrates. The samples were dried in a desiccator with a N_2_ atmosphere for 2 days. The samples were studied using LiteScope AFM (Nenovision, Brno, Czech Republic) with Akiyama cantilevers (NV-A-Probe from (Nanosensors, Neuchatel, Switzerland), tip radius of 15 nm, cantilever stiffness of 5 N/m), placed inside ZEISS EVO 10 SEM (Zeiss, Oberkochen, Germany). The AFM measurements were performed simultaneously with the SEM measurements at an acceleration voltage of 10 kV and working distance of 9 mm in regimes of secondary electrons (SE) or high-definition backscattered electrons (HDBSD). 

CPEM correlation and analysis of the obtained SEM and AFM microscopic images were performed using Matlab R2014b software. Fine-tuning of the offset between AFM and SEM images and slight SEM image stretching relative to the AFM image was required for more precise image correlation.

### 2.7. Surface Plasmon Resonance (SPR)

Common C1 (BS100540, subsequently SPR sensor 1) and a bare Au SPR chip (BR100405, subsequently SPR sensor 2) purchased from Cytiva (Uppsala, Sweden) were used. The latter was incubated with a mixed 1 mM thiol/ethanolic mixture to form a self-assembled monolayer (SAM) consisting of MUA and MCH (in a 1:10 ratio) during 1 h incubation at an ambient temperature (AT) in the dark. Subsequently, SPR sensors were used to study the affinity interactions of the exosomes with antibodies and lectins. All SPR experiments were performed using a Biacore X100 instrument (GE Healthcare, Vienna, Austria). Biacore X100 control and evaluation software were used to collect and analyse the data from all SPR measurements. 

In the pilot experiments, three different blocking solutions, i.e., two different 0.1 M bovine serum albumins (BSA) and 0.1 M ethanolamine hydrochloride, were compared. In addition, two different running buffers, i.e., phosphate-buffered saline-containing detergent (HBS-P+, buffer 10×; BR1006-71) and 10 mM PBS pH 7.4 were compared. First, immobilisation pH scouting was performed to determine suitable coupling conditions for the immobilisation of exosomes or antibodies. The interaction between the exosomes and antibodies was studied using two different approaches. In the first approach, the SPR chip containing carboxylic groups was activated using EDC/NHS (ratio 1:1) chemistry for 420 s. The diluted exosomes were immobilised on the sensor chip surface for a 1080 s association phase into cell 1 and cell 2, respectively. Finally, the activated unoccupied carboxylic groups on the SPR chip were blocked with 0.1 M ethanolamine hydrochloride (contact time of 420 s). The single cycle kinetics (SCK) approach was used to evaluate the interaction between the ligand immobilised on the chip surface (exosomes) and antibodies. The association and dissociation phases were set to 120 s and 60 s, respectively. Antibodies were diluted in 10 mM PBS pH 7.4 to attain concentrations of 6.725, 12.5, 25, 50, and 100 nM. The regeneration of the SPR sensor after all cycles was performed using 10 mM glycine/HCl pH 2.5 with a 60 s pulse and, finally, the chip was re-equilibrated with PBS. In the second approach, the antibodies were diluted in the acetate buffer pH 4.0 and immobilised on the sensor chip surface for 600 s into cell 1 and cell 2, respectively. The SCK procedure was chosen as previously described for the interaction examination between the ligand immobilised on the chip surface (antibodies) and exosomes. The extracted exosomes were diluted 22×, 44×, 88×, 176× and 352× in the PBS.

For the investigation of the exosome–lectin interaction, all SPR parameters were maintained as in the previous experiment. Lectins were diluted in PBS to final concentrations of 18.8, 37.5, 75, 150 and 300 nM (slightly higher than the antibodies due to higher K_D_). The appropriate glycoprotein-eluting solution (Vector Laboratories, Newark, CA, USA) described above was used for the sensor regeneration. The chip was finally re-equilibrated with a PBS running buffer. The sandwich configuration was achieved on both the SPR chips, the C1 chip and the mixed SAM-modified Au chip. First, the antibody anti-CD63 was immobilised onto the sensor surface and then diluted 2v1-derived exosomes were injected over the attached antibodies. In addition, the binding of the SNA lectin to the captured exosomes could be monitored using SCK. The SNA lectin was diluted as in the above-described experiments and the chip regenerated using a suitable glycoprotein-eluting solution for 60 s. For every SPR experiment, each chip was used (regenerated), at the most, three different times.

### 2.8. Electrochemical Measurements

Electrochemical measurements were performed using an Autolab analyser PGSTAT30 (Metrohm, Utrecht, The Netherlands) connected to a VA-Stand 663 (Metrohm, Herisau, Switzerland) with a three-electrode system. A hanging mercury drop electrode (HMDE, *A* = 0.4 mm^2^) as a working electrode, Ag/AgCl/3 M KCl as a reference electrode and Pt wire as an auxiliary electrode were used in a common thermostated cell open to the air. The working HMDE electrode was modified by exosomes using 5 min passive adsorption at open circuit potential, OCP. Then the exosome-modified HMDE was washed and transferred into a blank background electrolyte (50 mM Na-phosphate, pH 7) followed by C-E or C-t measurement. The analyte-modified electrode (modified by passive adsorption at open-circuit potential, OCP) was washed and transferred into a blank background electrolyte, followed by *C*-*E* curve measurement using FRA 2 software [32].

## 3. Results

### 3.1. NTA Characterisation

The exosomes isolated from both cell line media, i.e., cancerous 22Rv1 (carcinoma) and benign RWPE1 (control), were characterised by NTA. Single, well-defined symmetrical peaks were observed in NTA measurements for both samples (Figure 1). A clear difference can be seen in their concentration (~3× higher in the case of the carcinoma cell line) and an obvious difference in their mean hydrodynamic diameter (~9 nm difference, the carcinoma line being the smaller). Higher heterogeneity can also be observed for the control exosome population. The main peaks are highlighted between two grey zones where other fragments could also be visualised. For the 22Rv1-derived exosomes, a more detailed analysis revealed the presence of a high and sharp peak at 130.5 nm with a concentration of 4.13 × 10^6^ exosomes *per* mL. This was the main peak representing 92.1% of the integrated area of all the peaks present (minor peaks occurred at 225.5, 289.5 and 353.5 nm) (Figure 1). For the RWPE1-derived exosomes, detailed analysis revealed the presence of a high and sharp peak at 139.5 nm (slightly larger in diameter than the 22Rv1-derived exosomes) with a concentration of 1.39 × 10^6^ exosomes *per* mL. This was the main peak representing 60.8% of the integrated area of all the peaks present (minor peaks occurred at 82.5, 97.5, 198.5, 224.5 and 264.5 nm) (Figure 1). The size of the exosomes isolated from both the benign and cancerous cell lines is in excellent agreement with the values published elsewhere [33,34]. The concentration of cancerous and benign exosomes *per* mL is of the same order of magnitude (10^6^ exosomes *per* mL) as described previously for another prostate cell line LNCaP and benign RWPE1 cell line [35]. The number of exosomes produced by the benign RWPE1 prostatic cell line is lower than the 22Rv1-derived exosomes by a factor of ~3. This is in good agreement with the results obtained in our previous study, in which the kinetics of exosome production by benign and cancerous prostate cell lines was monitored [15].

### 3.2. DLS and ζ-Potential Characterisation

The size of the RWPE1-derived and 22Rv1-derived exosomes buffered by the PB-HAT pH 7.0 solvent was examined by a complementary DLS technique. A good correlation for the size of the exosomes’ particles was achieved through analysis by applying both techniques, i.e., NTA and DLS. The DLS analysis showed a peak at 132.9 nm for the observed size of 22Rv1-derived exosomes. The RWPE1-derived exosomes exhibited a peak size of 175.6 nm. Zeta-(*ζ*)-potential measurements yielded values of −14.9 mV and −9.9 mV for the carcinoma and control exosomes, respectively, suggesting a higher colloidal stability for the carcinoma exosomes due to a higher negative charge.

### 3.3. Microscopic Characterisation of Isolated Exosomes

In spite of the quite high dilution of exosomes for making samples for the microscopic measurements, they still had a tendency to make dense aggregated layers on the surface. The combined AFM-in-SEM technology [31] was used to find a suitable region of interest (ROI) for measurements of the exosome morphologies. In this method, SEM at various magnifications was used for navigation and finding ROI, and then the electron beam was focused near the AFM tip apex and by the sample movements beneath the AFM topography (Figure 2C,F) and SE intensity data (Figure 2D,G) were acquired simultaneously from the same ROI in the same conditions. HDBSD data are similar to SE data in terms of exosome size and shape, yet intensity distribution is more homogeneous as the topography has less impact on HDBSD data than on SE data. Both RWPE1-derived and 22Rv1-derived exosomes have rounded shapes with a small variation in size. The average size of both types of exosomes was ~150 nm in diameter (Figure 2A,B). This exosome size obtained by applying both microscopic techniques correlates well with the NTA results. Yet there are also noticeable larger exosomes (~300 nm) [33,34], which have visible depressions in the middle based on the AFM topography data. This could be due to a deformation of larger exosomes caused by the AFM probe; however, similar darker areas are also visible in the SEM images and such an effect can thus be excluded. Other possible reasons for such deformation of exosomes could be related to exosome drying (larger exosomes have a less rigid shape than the smaller exosomes). 

Figure 2E,H show the correlation analysis of AFM and SEM data, and in both cases, there is an overall positive correlation, i.e., the higher the object, the higher the electron emission from it [36]. By the correlation of SEM and AFM microscopic images, we were able to create a 3D correlative probe and electron microscopy (3D CPEM) view of the exosomes (Figure 2I,J). One can again see a good correlation between the topography and electron emission data. 

### 3.4. SPR Glycoprofiling

#### 3.4.1. Antibody Selection

Initially, immobilisation of the exosomes was optimised *prior* to SPR glycoprofiling. As was confirmed in our previous study, the CM5 SPR chip is not suitable for the immobilisation of exosomes [15]. We tested two different SPR chips with the intention of stable attachment of exosomes on the sensor surface. A commercially available sensor chip with a carboxyl terminated surface (C1—planar carboxylated surface, Cytiva; subsequently SPR sensor 1) was used in this study since it is recommended for use where the interaction partner in solution is multivalent or very large. This sensor chip keeps large particles as close to the surface as possible and reduces the avidity effect with multivalent interaction binding partners [37]. Before immobilising the exosomes, the pH values of the cell culture media and elution buffer were measured. Both the cell culture media had an average pH value of 7.2 shortly *prior* to exosome isolation. The elution and isolation buffer used for the exosomes’ collection as a part of the isolation kit had a pH value of 7.3. Based on these findings, PBS with a pH value of 7.4 was selected as the most convenient running and immobilising buffer—with a pH as close to physiological pH as possible. The average response value of 88 ± 31 pg·mm^−2^ for the immobilised exosomes was attained at the SPR sensor 1. When an HBS-P+ solution was used as a running buffer, the immobilisation of exosomes was unsuccessful and a low response of approximately 8 pg·mm^−2^ was achieved. An acceptable response and stability could only be achieved with PBS applied as a running and immobilisation buffer. The second chip to be tested (subsequently SPR sensor 2) demonstrated an average response of 957 ± 176 pg·mm^−2^ for the exosome immobilisation. We observed better reproducibility by employing the SIA Au sensor chip (modified by a mixed SAM consisting of MUA and MCH) with an RSD value of 18% compared with the C1 commercial chip with an RSD value of 36%. The subsequent SPR experiments using both types of sensors led to the investigation of exosome glycoprofiling and the exosome–antibody interaction. 22Rv1-derived exosomes were first immobilised on SPR sensor 1 and their affinity interactions with selected antibodies at five different concentration levels (6.25, 12.5, 25, 50 and 100 nM) were investigated. The results indicated that anti-CD63, anti-CD81 and anti-PSMA monoclonal antibodies were able to bind with their specific domains onto immobilised exosomes (Figure 3); however, anti-CD63 (orange line) yielded the best analytical sensitivity (slope of the calibration curve of RU signal at the end of a dissociation phase vs. concentration injected, fitted by a straight line) and LoD down to 0.22 nM (calculated as [3.3×σ]/slope). 

The above configuration (exosomes immobilised on the surface of SPR sensor 1 to achieve a final signal of 136 RU) yielded very similar results in the case of a reversed configuration, i.e., with antibodies immobilised on the chip surface. The K_D_ constant values measured for the anti-CD63, anti-CD81 and anti-PSMA antibodies were 1.9 × 10^−10^, 4.6 × 10^−9^ and 3.7 × 10^−10^ M, respectively. The most effective antibody to capture prostate cancer-derived exosomes was anti-CD63; hence, this was used in further experiments. 22Rv1 cell line-derived exosomes were used in these optimisation procedures as much higher quantities of the exosomes were available, and also due to the expectation that higher quantities of sialic acid-containing glycan epitopes should be present on these, higher initial signals could be achieved. This was further confirmed by applying different lectins using the same configuration shown in Figure 3, i.e., 22Rv1-derived exosomes were immobilised on the SPR sensor 1 chip surface, and different lectins were used as an analyte for kinetic titrations using five different concentrations, starting at 300 nM and further diluted by a factor of 2. 

#### 3.4.2. Lectin-Based In-Situ Glycoprofiling

A study of lectins revealed that the 22Rv1-derived exosomes related to prostate cancer mainly interacted with SNA and MAAII lectins. These lectins bind only specific carbohydrate structures with (*α*-2,6) linked and (*α*-2,3) linked sialic acid, with this also being the only monosaccharide carrying a negative charge, which is in good correlation with *ζ*-potential measurements. Detailed analysis revealed that other lectins were only able to bind poorly to carcinoma exosomes, i.e., PHA-E (galactose-, bisecting GlcNAc- and complex bi-antennary structures-binding), PSA (mannose- and glucose-binding) and WFL (*N*-acetylgalactosamine- and LacdiNAc-binding) [38,39]. Other investigated lectins yielded signals not significantly different from background noise. Mutual interactions were recorded on SPR sensor 1 with immobilised exosomes used as a ligand. The 22Rv1-derived exosomes immobilised on the sensor displayed an average response of 127 pg·mm^−2^, while control RWPE1-derived exosomes immobilised on the SPR sensor 1 exhibited an average response, which was 59% lower (i.e., 51.5 pg·mm^−2^). This correlates well with the other results. The 22Rv1-derived exosomes are slightly smaller than the control; hence, they could be immobilised on the surface of the sensor at a higher coverage density. The interaction ability of the lectins studied is summarised in Table 1.

The control exosomes mainly interacted with SNA and MAAII lectins; however, they exhibited a lower affinity than the carcinoma exosomes. Lectin-glycoprofiling showed that AAL, PHA-L and WFL were only able to bind poorly to RWPE1-derived exosomes, while there were no interactions between these lectins and carcinoma exosomes. Additionally, there were no interactions between the control exosomes and the GNL and PSA lectins, respectively (Table 1). One additional difference between the carcinoma and control exosomes was noted in the use of the PHA-E lectin, which binds to the control exosomes to a statistically higher extent than to carcinoma exosomes, yielding a signal difference of ~27 RU. The PHA-L lectin also binds to the control exosomes to a higher extent than to the carcinoma exosomes (ΔRU = 6.6). AAL, GNL, PSA and WFL yielded no difference at all. Comparing the absolute signal response at the end of a dissociation phase and analytical sensitivity between the carcinoma and control exosomes using SNA and MAAII lectins (both sialic acid-specific), the carcinoma exosomes yielded higher parameters and, hence (due to the fact that the same exosome concentration was used for the experiments), are more likely to contain higher amounts of surface sialic acid-containing epitopes, which, again, is in good correlation with the literature data (Figure 4) [40,41,42]. Consistent results could also be obtained for multi-cycle kinetics (MCK) analysis.

For each scenario, i.e., SNA being an analyte in the solution or immobilised on the chip surface, in the case of SPR sensors 1 and 2, this lectin always yielded the highest signals; hence, it was used for glycoprofiling in a sandwich configuration (Figure 5).

#### 3.4.3. SPR Sandwich Configuration

The SPR sandwich configuration is shown in Figure 5A. The interaction of exosomes and analytes was studied on both types of sensors. The response value of 838 pg·mm^−2^ for the immobilised anti-CD63 antibody was attained at SPR sensor 1. The higher response value of 2180 pg·mm^−2^ for the immobilised antibody was obtained using SPR sensor 2. In addition, the binding of the SNA lectin to the captured exosomes was examined. For the evaluation, the mutual interaction of the immobilised antibody as a ligand and SNA lectin as an analyte was measured and the response was subtracted from that obtained for the sandwich configuration, as IgG antibodies contain two bi-antennary *N*-glycans containing sialic acid at Asn297 in their Fc domain, possibly causing a cross-reaction with the lectin used and thus enhancing the background signal. The SPR sandwich formation was successfully proven with both types of sensors. Detailed analysis revealed a higher interaction of the attached carcinoma exosomes and the SNA lectin on SPR sensor 2. The summary of both experiments is shown in Figure 5B. A comparison of SCK for both types of SPR sensors is shown in Figure 5C,D.

### 3.5. Electrochemical Measurements

Stock solutions of exosomes from both cell lines (based on NTA data diluted to the same concentration by 10 mM PB pH 7.4 and stored at −80 °C) were thawed on ice *prior* to all measurements to obtain a fresh biological sample. Solutions were diluted twice, and the exosomes were allowed to be adsorbed on the surface of a mercury electrode at open current potential for 2 min. The dependence of capacitance on applied potential (*C*-*E* curves) was measured after washing and transferring the exosomes’ modified electrode to a measuring cell. High exosome concentrations were used to mitigate a possible effect of partial surface coverage. Figure 6 shows that the *C*-*E* curves for both types of exosomes (different cell lines) differed at potentials more positive than −0.7 V (Figure 6A). At *E* = −0.6 V, a reorientation of proteins occurs due to Hg-S bond reduction [43], while at potentials more negative than approximately −1.25 V, the exosomes are not in contact with HMDE, as the curves overlap with the curve of the background electrolyte [32]. The dependences of capacitance on time were performed, too, at slightly positively (−0.3 V; Figure 6C) and slightly negatively (−0.9 V; Figure 6B) charged electrodes according to the potential of zero charge (−0.4 V) to follow differences in adsorption kinetics. These time-dependent (*C*-*t*) curves were measured directly in exosome solution—a new drop was formed, and, at the same time, *C*-*t* curve reading was started. These curves thus represent the kinetics of exosome adsorption within a fixed timeframe. Both curves were fitted by an exponential fit model (the equation is shown in the graph inset), where the R0 parameter was used to quantify the rate of monolayer formation. 

The adsorption kinetics at two different potentials were also observed. In both cases, the 22Rv1 cells-produced exosomes’ monolayer is formed at a higher rate than for the control one. This can be attributed to the fact that the 22Rv1 exosomes are smaller and thus diffuse faster to the electrode surface (as shown by NTA and AFM). The rate constant ratio for the positively charged electrode *k*_22Rv1_/*k*_RWPE_ is 1.32—slightly higher than for the negatively charged electrode (1.22) suggesting the more rapid formation of the 22Rv1 exosomes’ layer due to electrostatic attraction between the positively charged electrode with the hypersialylated, i.e., negatively charged, surface of 22Rv1 exosomes. This result is in excellent correlation with the *ζ*-potential and SPR measurements shown above.

## 4. Discussion

The negative charge carried by sialic acids (so-called sialome) on the surface of cells contributes to the surface’s hydrophilicity and affects the cellular adhesion and metastatic potential of cancer cells [44]. Exosomes as small extracellular vesicles are known not only to modulate the primary tumour site but also to interact with a pre-metastatic niche [45]. The upregulation of sialyltransferases and the subsequent hypersialylation of various biomolecules attached to biological interfaces are hallmarks of many types of cancer, including lung, pancreatic, ovarian, breast and prostate [40,46]. Also, the role of aberrant sialic acid conjugates is linked to immune evasion, a process that has long been known and is crucial for any cell to maintain its malignancy [47]. Although discussed for many years, it is still not absolutely clear what the actual size of the exosomes is, with the literature recording a high variability ranging from 30–40 nm to 100, 150 or even 200 nm [12,48,49], all of which are values overlapping with the size of microvesicles (100–1000 nm, being directly shed from plasma membrane). In some cases, exosomes are not distinguished from other extracellular vesicles (EVs) with a diameter up to 250 nm, which would certainly contribute to greater misunderstanding and less coherent data in the research on exosomes in the event that methods other than affinity-based methods are to be used in the preparation of exosomes [50]. For this reason, isolation using the highly specific anti-CD63 antibody (targeting the CD63 tetraspanin involved in the formation of an early endosome) was chosen for exosome isolation in this study in order to mitigate any biological variability in the sample obtained by, for example, ultracentrifugation [51]. Recently, evidence of the different sizes of exosomes varying based on the state of producing cells has been published. For prostate cancer- and healthy individuals-derived exosomes isolated from real human samples (*n* = 107), their size has been measured as 131.4 nm and 145.9 nm, respectively (*p*-value < 0.05), using NTA [52]. Our NTA, DLS and AFM measurements are in excellent agreement with the results published by Logozzi et al. [27,28], confirming a slightly lower diameter of the measured particles isolated by affinity purification. For DLS measurements, however, we found that the use of PB as a storing solution for the purified exosomes is not optimal in cases where there is no flow in the system (as in the case of NTA or SPR systems). PB-HAT (PB + 20 mM HEPES + 0.2% Human Serum Albumin + 25 mM trehalose) stabilises the sample and makes the results much more reliable and similar to what the NTA measurements revealed, possibly preventing a dispersion from aggregation. This observation has also been made by Görgens et al. [25]. The higher aggregation rate could be anticipated for the control exosomes, as these carry less of a negative charge (as revealed by *ζ*-potential measurements); hence, they have lower colloidal stability, which again correlates well with the literature data [53]. Higher amounts of negative charge on the relatively smaller surface area of the carcinoma exosomes mean a higher charge density and thus higher stability in a solution. Any additives, such as trehalose and other saccharides, need to be chosen wisely in cases where the exosomes act as a binding partner for any lectins.

Sialic acid is not, however, the only carrier of negative charge on glycan molecules. Sulphated glycans are also a contributor to cancer metastatic and invasive potential [54]. Commercially available lectins do not usually bind to these structures; hence, in this study, only sialic acids as negative charge carriers were considered, although there are numerous proteins binding to these structures, e.g., heparin interactome or lysozyme binding to heparan sulphate [55,56]. Sialic acid, more specifically its higher content in cancer cells due to the overexpression of several sialyltransferases as well as other effects, and the occurrence of less common glycosidic bonds such as *α*-2,3 and *α*-2,8 (polysialic acid epitopes), are known indicators of cancer development and can be associated with the prognosis of different oncological diseases [57,58]. During SPR experiments, overall sialic acid content (SNA + MAAII) and both sialic acid-glycoforms (i.e., *α*-2,3 and *α*-2,6) individually were significantly increased in the case of 22Rv1 cell line-derived exosomes (carcinoma cell line), suggesting these post-translational modifications of exosome surface glycoconjugates represent a promising tool for liquid biopsy approaches, as these structures can be collected in a non-invasive way from different body fluids. In the case of prostate cancer, prostate-derived exosomes are contained in serum [59] and also in urine [60,61]. The first commercially available product on the market based on exosomes (IntelliScore by ExoDX) for the risk assessment of prostate cancer *prior* to initial prostate biopsy has already obtained its CE-IVD mark [62].

The negative charge on the exosome surface has also been used for their electrostatic-driven adsorption on the surface of a positively charged Hg electrode (HMDE) with an atomically smooth surface and high reproducibility during measurements. Both types of exosomes were adsorbed on the electrode interface (in the *C*-*E* curve, the signal was below that of a bare electrolyte solution); however, the adsorption disappeared almost completely at highly negative potentials, achieving the same signal as the background electrolyte. In the case of the positively charged Hg electrode (at *E* = −0.3 V vs. Ag/AgCl), adsorption of more negatively charged carcinoma exosomes was observed at rates higher than for control exosomes. In cases when the electrode was polarised to more negative potentials (at *E* = −0.9 V vs. Ag/AgCl), this difference in adsorption kinetics diminished and the minor difference was the result of the faster diffusion of smaller 22RV1 exosomes. 

Carcinoma and control exosomes can be distinguished not only by their charge and content of sialic acid epitopes, as we show results from the other lectins involved in this study, namely *Phaseolus vulgaris* lectin isoforms PHA-E and PHA-L. In the case of PHA-L, these are complex branched structures (multi-antennary *N*-glycans), which would normally be terminated by sialic acid and thus would increase the overall content of negative charge on the surface. The fact that the trend is reversed here, i.e., both the PHA lectins are binding more strongly to the control exosomes, may be caused by the fact that the increased sialylation present in the carcinoma exosomes negatively affects the PHA lectins binding to the underlying glycan structures (*β*-1,6-GlcNAc moiety from the glycan core or bisecting GlcNAc) or by the presence of polysialic acid not recognised by the SNA or MAAII lectin, but significantly affecting the availability of the glycans for biorecognition, especially since increased branching has also been linked to the process of tumourigenesis in the past [63]. There is strong evidence that the analyses of glycans and exosomes have great potential for being used in the future as a novel liquid biopsy approach. There is, however, also a need for an easy, cost-effective, reproducible and reliable method for their isolation and detection/characterisation. As we showed in this and a previous study [15], the surface morphology and electrostatic forces possibly present on bio-recognition interfaces needed to be considered for a sandwich format of analysis. For that purpose, sophisticated methods based on the integration of Au@SiO_2_/Au film metasurface and other nanoparticle-based surfaces [64,65] could be used to reduce the high variation in surface preparation, which is needed for clinical practice.

## 5. Conclusions

The study provided a detailed investigation of differences in the size, concentration and charge of exosomes produced by benign and cancerous cell lines. We introduced a method for the glycoprofiling of exosomes using SPR and identified prospective receptors on the surface of exosomes, which could be used for the bioaffinity isolation of exosomes. The abundance of receptors decreased in the order of CD63, PMSA and CD81. The most effective antibody to capture prostate cancer-derived exosomes was anti-CD63 and, thus, it has been used in further experiments. In addition, several lectins were applied to glycoprofile exosomes using SPR showing that four lectins, in particular, could be used for exosome discrimination, i.e., SNA and MAAII (recognising carcinoma exosomes) and PHA-E and PHA-L (recognising control exosomes). We also showed that it was possible to perform measurements in a sandwich configuration, i.e., antibody/exosomes/lectin using the SPR assay format, which could also be extended to another assay format. Electrochemical assay results are consistent with DLS results confirming that carcinoma exosomes are more negatively charged than control exosomes and our SPR results indicate that, especially, the presence of sialic acid-containing glycans can explain such observations. The results indicate that the glycosylation of exosomes has the potential for use as novel liquid biopsy markers.

## Figures and Tables

**Figure 1 sensors-24-01128-f001:**
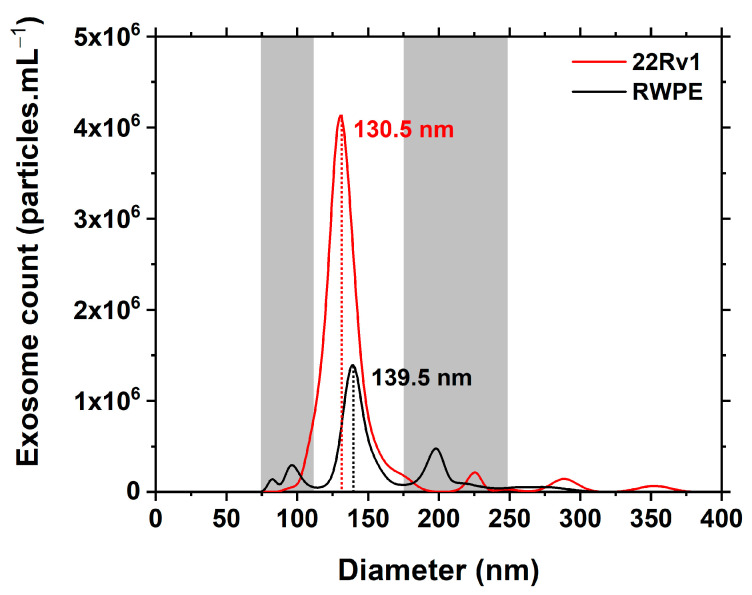
Representative graph of NTA analysis of particle size of 22Rv1-derived (red line) and RWPE1-derived (black line) exosomes.

**Figure 2 sensors-24-01128-f002:**
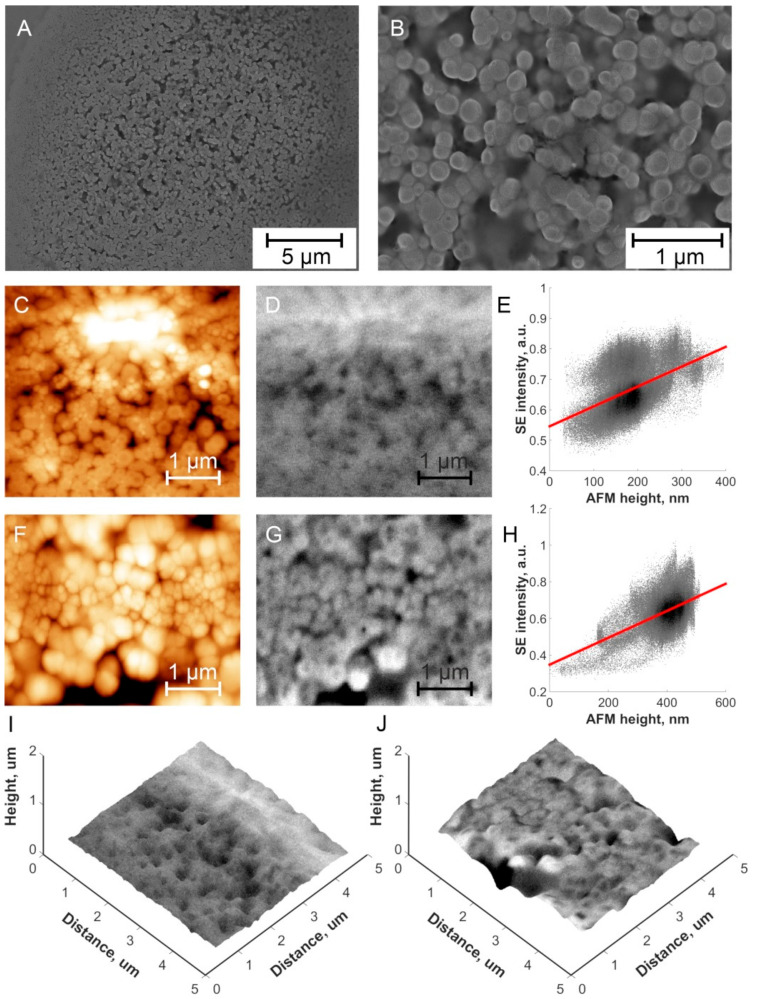
(**A**,**B**) SEM of Ca-derived exosomes, magnification 5000× (**A**) and 30,000× (**B**). (**C**) 2D topography AFM image of Ctr-derived exosomes. (**D**) 2D SE intensity SEM image of Ctr-derived exosomes. (**E**) SE intensity vs. AFM height correlation analysis for data obtained on Ctr-derived exosomes. (**F**) 2D topography AFM image of Ca-derived exosomes. (**G**) 2D SE intensity SEM image of Ca-derived exosomes. (**H**) SE intensity vs. AFM height correlation analysis for data obtained on Ca-derived exosomes. (**I**) 3D CPEM image of Ctr-derived exosomes, height data are taken from (**C**), colours from (**D**). (**J**) 3D CPEM image of Ca-derived exosomes, height data are taken from (**F**), colours from (**G**).

**Figure 3 sensors-24-01128-f003:**
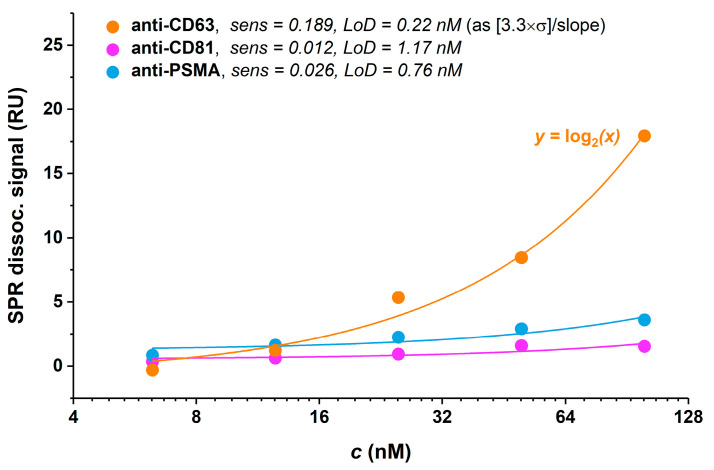
Calibration curves from SPR sensorgrams obtained via interaction of immobilised 22Rv1-derived exosomes with three different antibodies, viz., anti-CD63 (orange line), anti-CD81 (magenta line) and anti-PSMA (blue line) using SPR sensor 1. The analyte was injected into the system at concentrations of 6.25, 12.5, 25, 50 and 100 nM.

**Figure 4 sensors-24-01128-f004:**
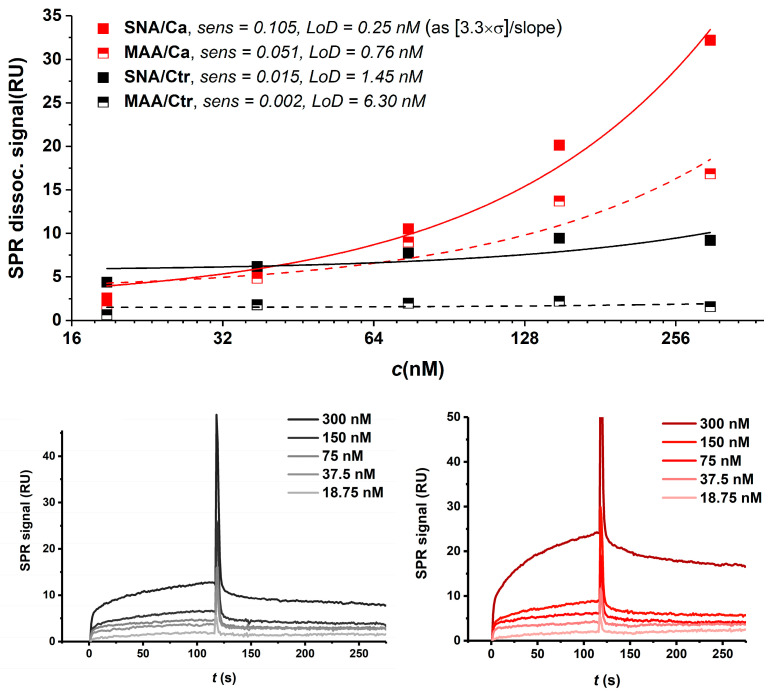
SPR sensorgram showing interaction of immobilised 22Rv1-derived exosomes with SNA and MAAII (both sialic acid-specific) lectins. Higher analytical sensitivities and lower limits of detection (LoDs) were achieved for carcinoma exosomes (**upper row**). SCK SPR sensorgrams showing interaction of SNA lectin with control (**lower left**) and carcinoma (**lower right**) exosomes. The analyte was injected into the system at concentrations of 18.75, 37.5, 75, 150 and 300 nM.

**Figure 5 sensors-24-01128-f005:**
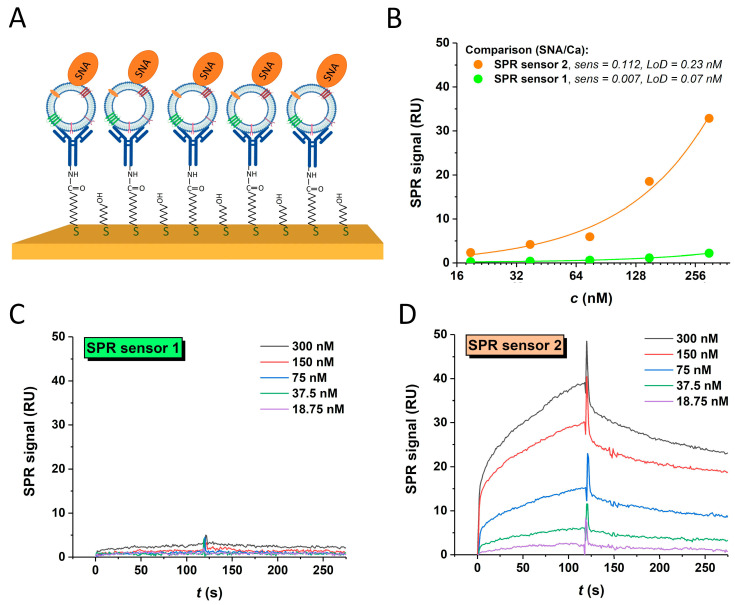
(**A**) Schematic presentation of sandwich configuration for exosome glycoprofiling (mixed SAM monolayer, i.e., SPR sensor 2, antibody—dark blue, lectin—orange) and calibration curves for both sensor types in sandwich configuration for in situ sialic acid detection on intact exosomes (**B**). SPR SCK procedure showing interaction of 22Rv1-derived exosomes, attached to immobilised anti-CD63 antibody, and SNA lectin for SPR sensor 1 (**C**) and SPR sensor 2 (**D**). The analyte was injected into the system at concentrations of 18.75, 37.5, 75, 150 and 300 nM.

**Figure 6 sensors-24-01128-f006:**
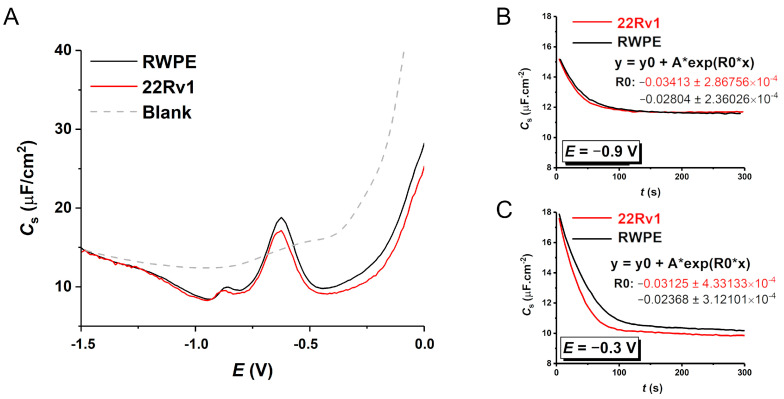
Electrochemical measurements of 22Rv1- (red line) and RWPE1-derived (black line) exosomes adsorbed ((**A**), *C*-*E* curve) and being adsorbed ((**B**,**C**), *C*-*t* curve, i.e., kinetic measurement) on the surface of HMDE. For a positively charged electrode, 22Rv1-exosomes are adsorbed at a higher rate.

**Table 1 sensors-24-01128-t001:** Summary of results obtained using lectins to bind (+) or not to bind (−) to exosomes immobilised at SPR sensor 1 chip.

Lectin/Agglutinin (Abbreviation)	Specificity	SU/GBS	Carcinoma Exosomes	Control Exosomes
*Aleuria aurantia* (AAL)	fucose, arabinose	2 subunits/5 fucose BS	−	−
*Galanthus nivalis* (GNL)	mannose	4 subunits/12 mannose BS	−	−
** *Maackia amurensis* ** **(MAAII)**	*α*-2,3 linked Sia	2 subunits	+	+
** *Phaseolus vulgaris* ** **(PHA-E)**	galactose, bisecting GlcNac, biantennary N-glycans	4 E subunits	−	+
** *Phaseolus vulgaris* ** **(PHA-L)**	branched N-glycans	4 L type subunits	−	+
*Pisum sativum* (PSA)	mannose, glucose	4 subunits	−	−
** *Sambucus nigra* ** **(SNA)**	*α*-2,6 linked Sia	2 subunits	+	+
*Wisteria floribunda* (WFL)	N-Acetylgalactosamine	4 subunits	−	−

Sia = sialic acid, SU = subunits, (G)BS = (glycan-)binding sites, bold lectins = lectins positively interacted with the exosomes.

## Data Availability

Data are available upon request.

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
