# Peer review of "Negative Charge-Carrying Glycans Attached to Exosomes as Novel Liquid Biopsy Marker"

_sensors, 2024, doi:10.3390/s24041128_

Round 1
Reviewer 1 Report
Comments and Suggestions for Authors
The paper presents valuable insights into the isolation and characterization of exosomes as potential liquid biopsy markers for early prostate cancer diagnosis. With its comprehensive experimental approach and significant findings, the article is suitable for publication. Minor revisions are recommended:
1. Ensure that the first bracket in line 28 is properly closed.
2. Correct the writing errors in line 200 ("sample The samples") and line 274 ("Is is seen").
3. Could you please specify the concentration range used for the calibration in Figure 3, as the sensitivity is determined by the slope of the calibration curve of the RU signal?
4. I suggest replacing the concentration labels 37,5 nM and 18,75 nM in Figures 4 and 5 with the more conventional notation 37.5 nM and 18.75 nM.
5. There appears to be a discrepancy between the statement in line 472, "slightly positively (-0.3 V; Fig. 6B) and slightly negatively (-0.9 V; Fig. 6C)," and the labeling of Figures 6B and C. Please correct the labeling to accurately represent the mentioned potential values.
Author Response
The response is in the file attached.

Reviewer 2 Report
Comments and Suggestions for Authors
The authors have reported on exosomes derived from prostate-cancer cells as a biomarker. This paper is topical and could be of interest to a wide range of researchers in the field. However, there are a few issues that need to be addressed before suggesting for publication.
The overall English would benefit from a proof reading.
The introduction needs a revision as some of the information i.e in the second paragraph is rather trivial. There is no connection between different paragraphs in the into. Also, the authors are suggested to provide a background information on the current state of the art in detection of exosomes as disease biomarker (e.g. doi.org/10.1002/anbr.202300055)
It is mentioned in the abstract that “The results indicate that cancerous exosomes are smaller, produced at higher concentrations and…” which is contradicting the following statement from results section “slight difference in their mean hydrodynamic diameter” were the size differences is only 9nm.
The scale bars in figure.2 are not legible.
Have the authors done any PCR or WB test to confirm the presence of these antibodies?
The acronyms for the two cells lines used in this paper are not consistent throughout the paper, especially in the figures.
Comments on the Quality of English Language
Minor editing of English language required
Author Response
The response is in the file attached.
